# Relationship between Novel Elastography Techniques and Renal Fibrosis—Preliminary Experience in Patients with Chronic Glomerulonephritis

**DOI:** 10.3390/biomedicines11020365

**Published:** 2023-01-26

**Authors:** Felix-Mihai Maralescu, Adrian Vaduva, Adalbert Schiller, Ligia Petrica, Ioan Sporea, Alina Popescu, Roxana Sirli, Alis Dema, Madalina Bodea, Iulia Grosu, Flaviu Bob

**Affiliations:** 1Division of Nephrology, Department of Internal Medicine II, Victor Babeș University of Medicine and Pharmacy, 300041 Timisoara, Romania; 2Centre for Molecular Research in Nephrology and Vascular Disease, Victor Babeș University of Medicine and Pharmacy, 300041 Timisoara, Romania; 3County Emergency Hospital, L. Rebreanu Street, Nr. 156, 300723 Timisoara, Romania; 4ANAPATMOL Research Centre, Discipline of Morphopathology, Department of Microscopic Morphology, Victor Babes University of Medicine and Pharmacy, 300041 Timisoara, Romania; 5Discipline of Morphopathology, Department of Microscopic Morphology, Faculty of Medicine, Victor Babes University of Medicine and Pharmacy, 300041 Timisoara, Romania; 6Department of Gastroenterology and Hepatology, Victor Babeș University of Medicine and Pharmacy, 300041 Timisoara, Romania; 7Advanced Regional Research Center in Gastroenterology and Hepatology, Victor Babeș University of Medicine and Pharmacy, 300041 Timisoara, Romania

**Keywords:** renal biopsy, fibrosis, inflammation, stiffness, viscosity

## Abstract

Introduction: A renal biopsy represents the gold standard in the diagnosis, prognosis, and management of patients with glomerulonephritis. So far, non-invasive elastographic techniques have not confirmed their utility in replacing a biopsy; however, the new and improved software from Hologic Supersonic Mach 30 is a promising method for assessing the renal tissue’s stiffness and viscosity. We investigated whether this elastography technique could reveal renal tissue fibrosis in patients with chronic glomerulonephritis. Materials and methods: Two-dimensional-shear wave elastography (SWE) PLUS and viscosity plane-wave ultrasound (Vi PLUS) assessments were performed in 40 patients with chronic glomerulopathies before being referred for a renal biopsy. For each kidney, the mean values of five stiffness and viscosity measures were compared with the demographic, biological, and histopathological parameters of the patients. Results: In total, 26 men and 14 women with a mean age of 52.35 ± 15.54 years, a mean estimated glomerular filtration rate (eGFR) of 53.8 ± 35.49 mL/min/1.73m^2^, and a mean proteinuria of 6.39 ± 7.42 g/24 h were included after providing their informed consent. Out of 40 kidney biopsies, 2 were uninterpretable with inappropriate material and were divided into four subgroups based on their fibrosis percentage. Even though these elastography techniques were unable to differentiate between separate fibrosis stages, when predicting between the fibrosis and no-fibrosis group, we found a cut-off value of <20.77 kPa with the area under the curve (AUC) of 0.860, a *p* < 0.001 with 88.89% sensitivity, and a 75% specificity for the 2D SWE PLUS measures and a cut-off value of <2.8 Pa.s with an AUC of 0.792, a *p* < 0.001 with 94% sensitivity, and a 60% specificity for the Vi PLUS measures. We also found a cut-off value of <19.75 kPa for the 2D SWE PLUS measures (with an AUC of 0.789, *p* = 0.0001 with 100% sensitivity, and a 74.29% specificity) and a cut-off value of <1.28 Pa.s for the Vi PLUS measures (with an AUC 0.829, *p* = 0.0019 with 60% sensitivity, and a 94.29% specificity) differentiating between patients with over 40% fibrosis and those with under 40%. We also discovered a positive correlation between the glomerular filtration rate (eGFR) and 2D-SWE PLUS values (*r* = 0.7065, *p* < 0.0001) and Vi PLUS values (*r* = 0.3637, *p* < 0.0211). C reactive protein (CRP) correlates with the Vi PLUS measures (r = −0.3695, *p* = 0.0189) but not with the 2D SWE PLUS measures (*r* = −0.2431, *p* = 0.1306). Conclusion: Our findings indicate that this novel elastography method can distinguish between individuals with different stages of renal fibrosis, correlate with the renal function and inflammation, and are easy to use and reproducible, but further research is needed for them to be employed routinely in clinical practice.

## 1. Introduction

CKD has become a global public health problem with an increasing incidence and prevalence, resulting in a poor quality of life and high healthcare costs, and is linked to significant levels of mortality and cardiovascular morbidity, with leading causes such as diabetes mellitus, hypertension, and glomerulopathies [1,2,3,4].

The decline in the renal function is the consequence of tissue scarring and kidney parenchyma damage. Histopathology reveals that glomerulosclerosis, tubulointerstitial fibrosis, capillary loss, and tubular atrophy with the process of fibrosis are pathological conditions defined by the buildup and deposition of extracellular matrix constituents.

Despite the increasing use of kidney biomarkers, fibrosis is evaluated solely by a kidney biopsy and a histopathological diagnosis continues to be the most significant diagnostic and prognostic method for this chronic renal illness. On the other hand, acute inflammatory activation is characterized by the abundance of neutrophils and macrophages, drawn and triggered by the cytokine production in injured tissue, which then stimulates the adaptive immune response; this aspect has also been cited as a crucial factor in the progression of fibrosis [5].

By establishing appropriate therapy, an early renal biopsy may minimize the progression of CKD and, consequently, death [6,7]. However, as with any invasive operation, a kidney biopsy may potentially be associated with complications, such as serious bleeding (1 in 1000 patients), the requirement for angiographic therapy (1 in 2000 patients), unilateral nephrectomy (1 in 10,000 patients), and even mortality (1 in 5000 patients); therefore, the advantages and drawbacks must always be weighed [8].

Recent years have witnessed the emergence of elastography as a non-invasive imagistic technique for identifying widespread disorders and this has boosted its usage in the medical community, particularly among gastroenterologists and endocrinologists [9,10]. The approach’s major objective for nephrologists would be to noninvasively diagnose fibrosis, inflammation, and monitor CKD with its progression over time [11,12].

Because serum creatinine levels and eGFR are poor indicators of the severity of histological abnormalities in kidneys, a noninvasive test, that might provide an early diagnosis and/or prognosis to minimize the need for kidney biopsies and perhaps allow for early, targeted therapy, could improve patient care and survival. Up until now, even though kidney elastography research revealed an increased heterogeneity [12], multiple kidney elastography studies indicate a clear relationship between renal stiffness and either fibrosis or the renal function [13,14,15,16,17,18]. However, to our knowledge, no other study has compared this new, improved renal elastography software which is able to measure at the same time both the tissue’s stiffness with 2D-shear wave elastography (2D-SWE PLUS) and a viscosity plane-wave ultrasound (Vi PLUS), with histopathology findings.

Therefore, the purpose of this study was to determine the viability of the ultrasound-based techniques provided by the Hologic Aixplorer Mach 30 system (Aixplorer, Supersonic Imagine, Aix-en-Provence, France) and compare the results with the “gold standard” of a kidney biopsy in patients with chronic glomerulonephritis.

## 2. Materials and Methods

### 2.1. Patient Selection

In a tertiary nephrology department over a ten-month period (March 2022 to December 2022), cross-sectional, monocentric research was undertaken. Forty patients with chronic glomerulonephritis, who underwent elastography measures and then performed a renal biopsy, were included. In the Department of Nephrology Timisoara, Romania, the participants provided their informed consent and the study was conducted in line with the Declaration of Helsinki and was approved by our university’s ethical committee for research and institutional review board (41/4 March 2022).

Out of each individual’s medical records, we collected the following information: their age, gender, height, weight, body mass index (BMI), kidney length, if they presented a history of hypertension or diabetes, as well as the following blood analysis: a complete blood count, urea, serum creatinine, sodium, potassium, uric acid, erythrocyte sedimentation rate (ESR), C reactive protein (CRP), cholesterol, triglycerides, aspartate aminotransferase (ASAT), alanine aminotransferase (ALAT), and total bilirubin. The patients were also directed to provide urine samples for proteinuria/within 24 h and a urine culture.

### 2.2. Elastography

Before being referred for a renal biopsy, a single operator with five years of experience in renal ultrasonography performed elastography-based measurements on all individuals during the same session (F.-M.M), using the renal software comprising both 2D-SWE PLUS and Vi PLUS from the new Hologic Aixplorer Mach 30 ultrasound system (Aixplorer, Supersonic Imagine, Aix-en-Provence, France).

For each separate kidney, consecutive measurements were performed in the central section of the renal parenchyma, right under the subcapsular cortex, with the patient in dorsal decubitus immediately after voiding the bladder. Five consecutive measures of renal stiffness and viscosity were obtained at the same time without the knowledge of the patient’s medical history. The EFSUMB guidelines provide no quality criteria when performing kidney elastography, but considering the results of the majority of the published studies, we decided that the kidney’s stiffness should be reported as the mean value of 5 valid measurements.

For each measurement (with the region of interest (ROI) selected by the renal software system to 10 mm and presented on the screen as a Q-box), the equipment software produced the following data, which are presented in Figure 1.

A further improvement has been achieved by analyzing the raw data obtained from the elastography devices, as proposed by R. Barr et al. in 2020, and the new software from Hologic Aixplorer Mach 30 has more accurate and reliable renal stiffness information and uses more advanced processing techniques, represented as the 2D-SWE PLUS measures in kilopascal (kPa) [19].

The 2D SWE PLUS measurements were carried out with a C6-1X convex transducer. The Young’s modulus (YM) of the ROI was calculated with the apparatus’ software utilizing the formula E = ρ × cs2 (E represents the tissue elasticity (in kPa), ρ is the tissue density (in kg/m3), and cs is the shear wave velocity measures in m/s). A quantitative map of the tissue’s stiffness is displayed using ultrafast imaging techniques, with a color scale spanning from dark blue to yellow then to dark red, corresponding to the YM values ranging from 0 to >50 kPa [20].

Another feature of the new ultrasound device is Vi PLUS, which allows users to receive information on the tissue shear wave dispersion, that may be used to deduce the viscosity [21]. Vi.PLUS enables the presenting of information on the tissue shear wave dispersion (study of the shear wave propagation velocity at many frequencies) and the magnitude of change in the shear wave speed across the frequencies is depicted intuitively in a color-coded graphic and quantitatively in Pascal-second (Pa.s), represented as a unit of the dynamic viscosity over a range of values. 

The kidneys’ stiffness would represent a non-invasive marker of fibrosis and the kidneys’ viscosity would represent a non-invasive marker of inflammation.

### 2.3. Renal Biopsy and Histopathology

After elastography, under ultrasonic guidance, a kidney biopsy was conducted in the right inferior pole of the kidney parenchyma with an 18G needle. The biopsy of the kidney tissue was collected and then fixed in 10% formalin before being sent for a histopathological investigation. Hematoxylin and eosin, periodic acid Schiff, Masson trichrome, Congo red, and silver stain were utilized to analyze the paraffin slices. The biopsy samples were sent to the same pathologist (A.V) with 10 years of experience to be examined by light microscopy; the pathologist was blinded to the clinical data of the patients. Based on the severity of interstitial fibrosis, the patients were divided into four categories: no/minimal fibrosis (0–10%) (Figure 2), mild fibrosis (10–30%) (Figure 3), moderate fibrosis (30–50%) (Figure 4), and severe fibrosis (>50%) (Figure 5).

The biopsies were then examined using a semi-quantitative methodology, as was proposed by the Consensus Definitions for Glomerular Lesions by Light and Electron Microscopy: Recommendations from Working Group of the Renal Pathology Society by Hass et al., 2020 [22] and the Revised ISN/RPS 2018 classification of lupus renal pathology by Krassanairawiwong et al., 2020 [23].

### 2.4. Statistical Analysis

For the statistical analysis, MedCalc Version 19.4 (MedCalc Software Corp., Brunswick, ME, USA) and Microsoft Office Excel 2019 were utilized (Microsoft for Windows). Using the Kolmogorov–Smirnov test, the distribution of the numerical variables was analyzed. Means and standard deviations are used to describe the variables with a normal distribution, whereas median values and ranges are used to represent the variables with a non-normal distribution. Percentages and numbers were utilized to illustrate the qualitative elements. Using the Pearson correlation coefficient, the relationships between the variables were expressed. A *p*-value of less than 0.05 was judged as being statistically significant. The areas under the receiver operating characteristic curves (AUC) were calculated for 2D SWE PLUS and Vi PLUS and offered as cut-off points that maximized the Youden index for recognizing fibrosis. Univariate and multivariate statistical analyses were also implied.

## 3. Results

A total of 26 men and 14 women with a mean age of 52.35 ± 15.54 years, a mean body mass index (BMI) of 26.71 ± 4.65, a mean kidney length of 104.33 ± 20.19 mm, a mean estimated glomerular filtration rate (eGFR) of 53.8 ± 35.49 mL/min/1.73 m^2^, and a mean proteinuria of 6.39 ± 7.42 g/24 h underwent elastography then renal biopsy procedures. No patient had a positive urine culture, but 8 had a history of diabetes, and 34 had a history of hypertension (Table 1).

The mean 2D-SWE PLUS value for the whole group was 23.8 ± 7.45 kPa and the mean Vi PLUS value was 2.39 ± 0.73 Pa.s at a mean depth of measurement of 6.22 ± 1.43 cm. (Figure 6 and Figure 7 show the relative frequency of the elastography measures). eGFR correlate with both 2D-SWE PLUS measures (r = 0.7065, *p* < 0.0001) and Vi PLUS measures (r = 0.3637, *p* = 0.0211).

Univariate regression shows that 2D-SWE PLUS is influenced by cholesterol (*p* = 0.0085), hemoglobin (*p* = 0.0001), eGFR (*p* < 0.0001), urea (*p* < 0.0001), and Vi PLUS (*p* < 0.0001). However, in a multivariate regression model, only eGFR (*p* < 000.1) and Vi PLUS (*p* < 0.001) were independently associated with the 2D SWE PLUS measures (*p* < 0.0001).

Univariate regression shows that Vi PLUS is influenced by cholesterol (*p* = 0.0020), CRP (*p* = 0.0189), eGFR (*p* = 0.0211), 2D-SWE PLUS (*p* < 0.0001), and urea (*p* = 0.0346). The multivariate regression model shows that only eGFR (*p* = 0.0067) and CRP (*p* = 0.0061) were independently associated with the Vi PLUS PLUS measures (*p* = 0.0016).

Out of the 40 biopsies, 2 were uninterpretable with inappropriate material, 5 patients presented with minimal change disease, 4 with lupus nephritis, 4 with membranous nephropathy, 7 with focal segmental glomerulosclerosis, 6 with rapidly progressive glomerulonephritis, 2 with amyloidosis, 2 with membranoproliferative glomerulonephritis, 4 with mesangial proliferative glomerulonephritis, and 4 with diabetic nephropathy (Figure 8).

Based on their interstitial fibrosis percentage, the patients were divided into four categories: no/minimal fibrosis (0–10%): 20 patients, mild fibrosis (10–30%): 12 patients, moderate fibrosis (30–50%): 4 patients, and severe fibrosis (>50%): 2 patients; 2 patients were uninterpretable (Figure 9).

Even though these elastography techniques were unable to differentiate between separate fibrosis stages, predicting between the fibrosis (over 10%) and no-fibrosis group (0–10%), we found a cut-off value of <20.77 kPa for detecting the presence of fibrosis with the area under the curve (AUC) of 0.860, *p* < 0.001 with 88.89% sensitivity, and a 75% specificity for the 2D SWE PLUS measures and a cut-off value of <2.8 Pa.s for detecting the presence of fibrosis with an AUC of 0.792, *p* < 0.001 with 94% sensitivity, and a 60% specificity for the Vi PLUS measures (Figure 10 and Figure 11).

We also found a cut-off value of <19.75 kPa for the 2D SWE PLUS measures (with an AUC of 0.789, *p* = 0.0001 with 100% sensitivity, and a 74.29% specificity) and a cut-off value of <1.28Pa.s for the Vi PLUS measures (with an AUC 0.829, *p* = 0.0019 with 60% sensitivity, and a 94.29% specificity) differentiating between patients with over 40% fibrosis and those with under 40% (Figure 12 and Figure 13).

## 4. Discussion

In reality, renal elastography will never be able to match the gold standard, which is represented by a renal biopsy, and is unlikely to surpass histology for defining fibrosis and a loss of the kidney function. However, the capability to track changes in the parenchymal structure over time would be the most feasible and attractive use of this method. Even if the serum creatinine levels remain stable, a progressive decrease in parenchymal stiffness and viscosity over consecutive 2D SWE PLUS and Vi PLUS measurements would provide nephrologists with a better understanding of the progression of fibrosis in the kidneys.

The decision to undergo a biopsy is sometimes declined by the patient because of the rare but possible complications that can occur. 2D SWE PLUS and Vi PLUS, on the contrary, are rapid, noninvasive procedures for measuring the kidney’s stiffness and viscosity and establishing the presence of fibrosis and inflammation, with a high patient acceptability, excellent repeatability, and immediate findings [18,24].

The mean 2D-SWE PLUS value for the whole group was 23.8 ± 7.45 kPa and the mean Vi PLUS value was 2.39 ± 0.73 Pa.s, which correlated with the renal function (eGFR), but neither of them correlated with age progression. In another study of our group in healthy participants, we found a mean kidney stiffness value of 31.88 ± 2.89 kPa and a mean viscosity value of 2.44 ± 0.57 Pa.s, and both correlated with eGFR and age [25]. A possible explanation would be that both the stiffness and viscosity of the kidney decrease as the CKD stages progresses. Unmistakably, age has an influence on renal rigidity because degeneration is a physiological phenomenon of cellular and organ aging and is thus associated with structural alterations in the kidneys. Due to these modifications, renal blood flow declines with age, which may be a possible reason for the decline in stiffness with increasing years [26].

Not all studies on kidney elastography demonstrate the same association between stiffness and histopathological alterations in the kidney. Most investigations comparing elastography with histological factors were conducted on patients with renal transplants [16,18,24,27,28,29,30]. Studies utilizing several elastography techniques on native kidneys demonstrate a statistically significant rise in the kidney stiffness in individuals with more advanced kidney histopathological abnormalities [13,31]. In a study also utilizing 2D-SWE elastography technology, it was discovered that in addition to kidney stiffness, which positively statistically correlates with the degree of glomerulosclerosis and tubulointerstitial fibrosis, individuals with less kidney stiffness also responded better to cortico-therapy treatment [32].

One research found no statistically significant association between kidney stiffness and the histopathological characteristics which were evaluated (glomerulosclerosis index, tubular atrophy, and interstitial fibrosis) [33]. Additionally, in a separate study involving living kidney transplant donors in whom elastography was performed prior to a surgical excision and renal biopsies were conducted prior to implantation, there were no statistically significant associations between the histological parameters and the kidneys’ stiffness [34].

Hu Q. et al., 2019 [35] found, by including 163 patients with CKD and 32 healthy individuals, that people with advanced glomerular, interstitial, tubular, and vascular lesions had decreased kidney stiffness. This aspect was also pinpointed in a smaller trial conducted on patients with glomerulonephritis, where the impact of tubulointerstitial fibrosis or arteriolar hyalinosis was linked to dramatically lower stiffness readings [36].

Our findings clearly demonstrate a connection between reduced cortical stiffness and viscosity and the presence of fibrosis. The vast majority of earlier studies on native kidneys employed acoustic radiation force impulse technology and found as well significantly lower stiffness values in the CKD population utilizing biopsy, GFR, serum creatinine, and scintigraphy as indicators for a compromised renal function [37,38].

The 2D SWE PLUS and Vi PLUS methods were initially developed to identify liver fibrosis and inflammation [39]; the present investigation proved that these techniques could also be used to identify renal fibrosis and inflammation. The most surprising observation we discovered was the correlation of CRP with mean viscosity values (r = −0.3695, P = 0.0189) and not with kidney stiffness values (r = −0.2431, P = 0.1306). It is true, however, that a patient with systemic lupus erythematosus, complicated diabetes mellitus, or even simply CKD may have elevated CRP for other reasons. In contrast with a previous study of ours in kidney transplanted patients [11], we found no correlation between the viscosity measures and CRP, and this may be attributable to immunosuppressive medication. Similarly, Sugimoto et al., 2018 [21] performed an elastography study in rat liver models for identifying the level of fibrosis, where elasticity exceeded the viscosity, but the viscosity excelled over the elasticity in identifying the amount of necroinflammation. Therefore, viscosity measures that highlight inflammatory states might be an advantageous alternative for evaluating patients with an acute kidney injury or with acute pyelonephritis, as well as when evaluating acute rejections in kidney transplants. Unfortunately, our investigation revealed no correlation between rapidly progressive glomerulonephritis and elevated viscosity levels.

Nevertheless, despite the fact that renal elastography might be a promising method for monitoring the course of CKD, the research so far indicates that its heterogeneity is developed [12]. The structure of the renal parenchyma, which is defined by a high extent of anisotropy, the profundity of the kidneys, and the thickness of the renal parenchyma, influences the observed kidney stiffness and the viscosity readings, making it challenging to interpret the findings.

Several limitations of our research should be acknowledged: the limited number of patients, the lack of an activity/chronicity index given the heterogenous etiologies of glomerulonephritis, the use of a standardized method, and quality criteria for performing elastography in kidneys.

## 5. Conclusions

Our findings indicate that these novel elastography methods can distinguish between individuals with different stages of renal fibrosis, correlate with the renal function and inflammation, and they are easy to use and are reproducible, with a high patient acceptance, but further research is needed for them to be employed routinely in clinical practice.

## Figures and Tables

**Figure 1 biomedicines-11-00365-f001:**
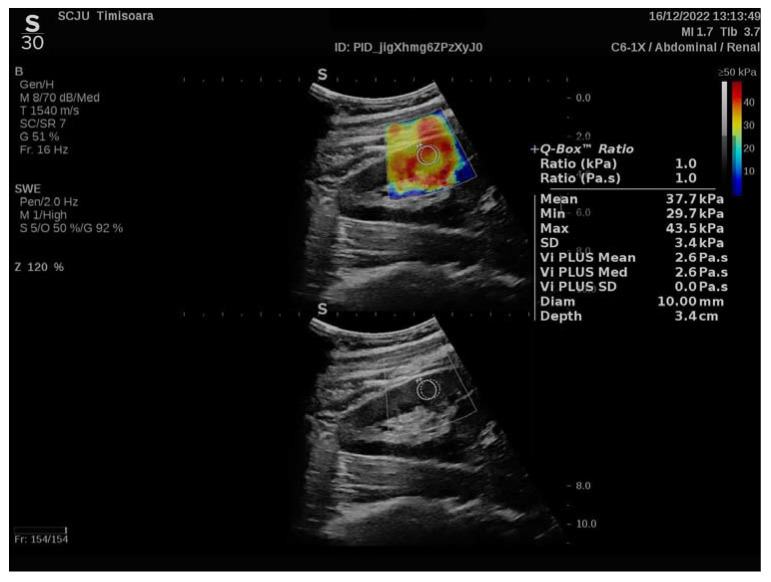
Elastography of a chronic glomerular patient’s kidney before being referred for renal biopsy.

**Figure 2 biomedicines-11-00365-f002:**
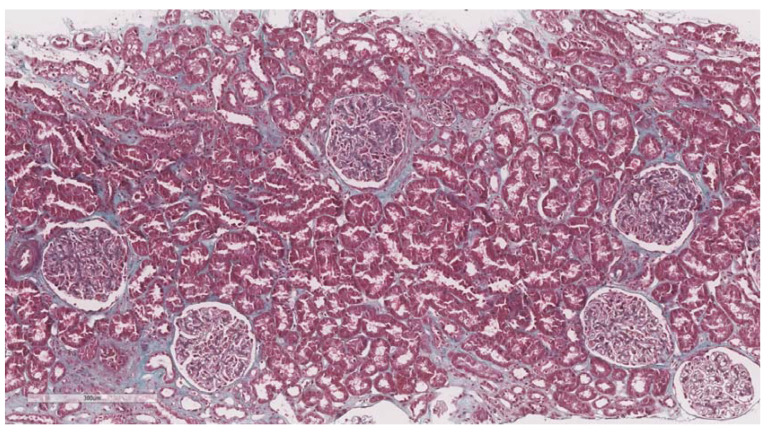
10× objective image of a kidney biopsy of a patient with no/minimal interstitial fibrosis (0–10%) evaluated on trichrome stain.

**Figure 3 biomedicines-11-00365-f003:**
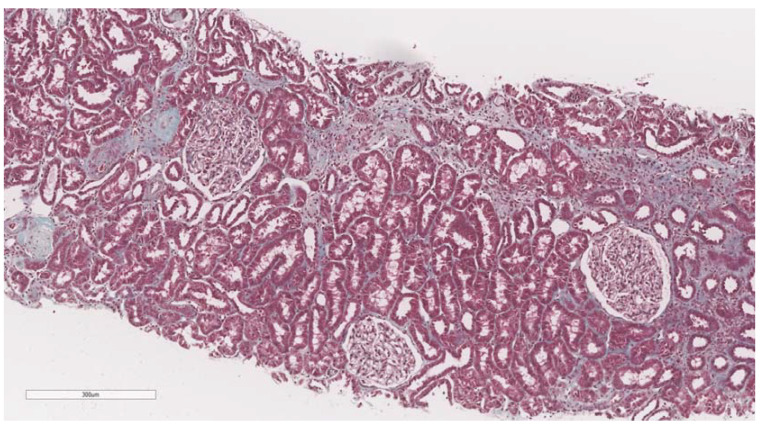
10× objective image of a kidney biopsy of a patient with mild interstitial fibrosis (10–30%) evaluated on trichrome stain.

**Figure 4 biomedicines-11-00365-f004:**
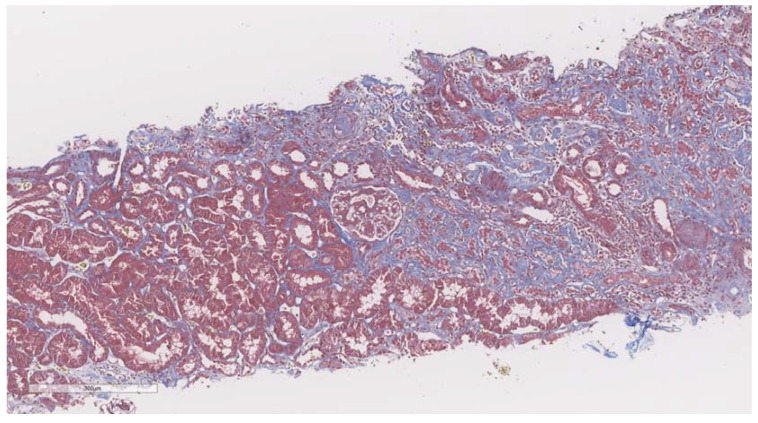
10× objective image of a kidney biopsy of a patient with moderate interstitial fibrosis (30–50%) evaluated on trichrome stain.

**Figure 5 biomedicines-11-00365-f005:**
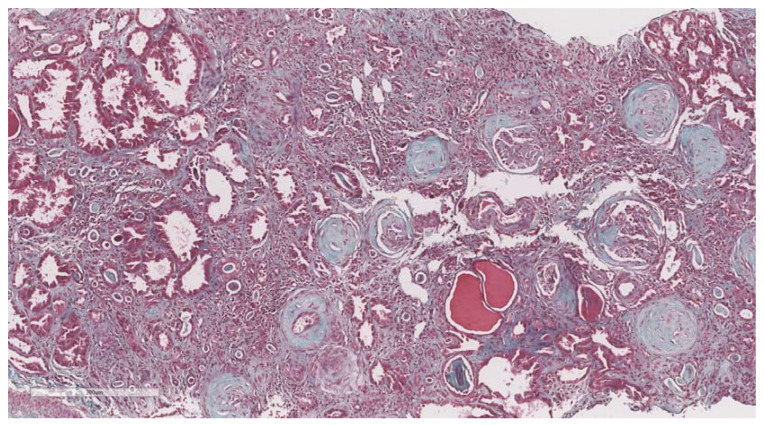
10× objective image of a kidney biopsy of a patient with severe interstitial fibrosis (>50%) evaluated on trichrome stain.

**Figure 6 biomedicines-11-00365-f006:**
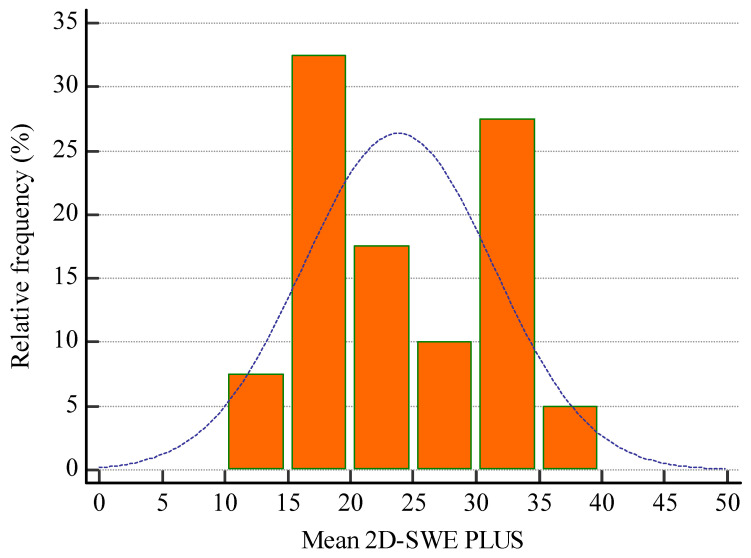
Relative frequency of mean 2D-SWE PLUS measures.

**Figure 7 biomedicines-11-00365-f007:**
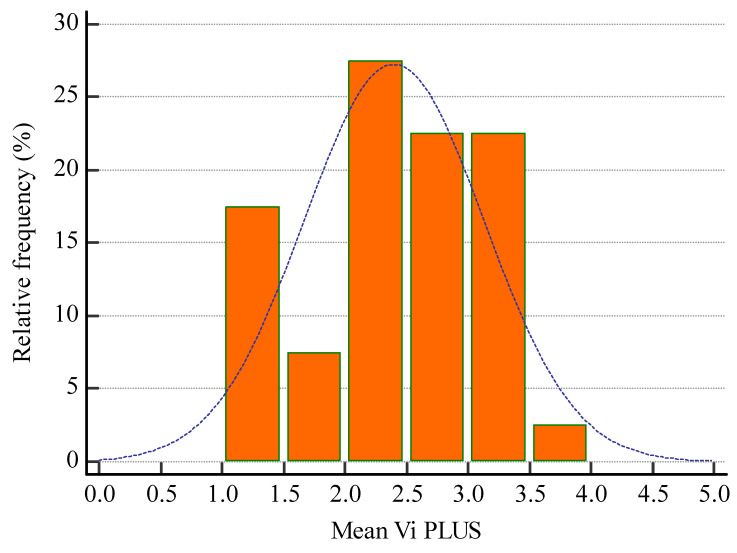
Relative frequency of mean Vi PLUS measures.

**Figure 8 biomedicines-11-00365-f008:**
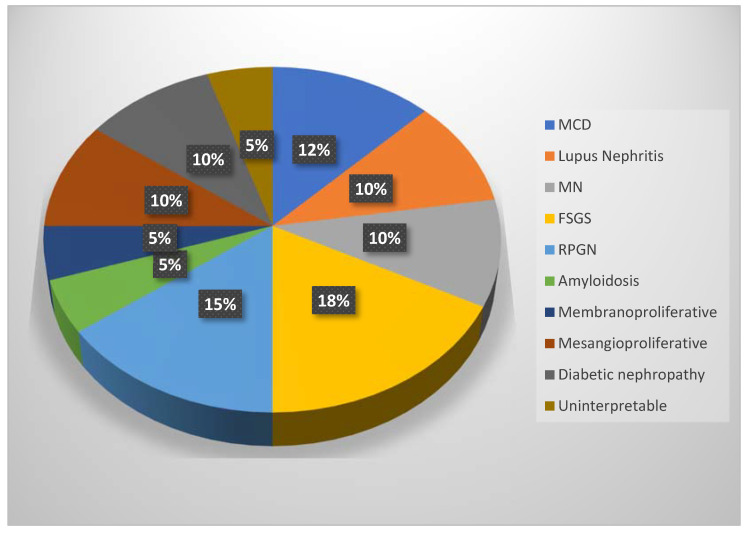
Pie chart for the types of glomerulonephritis (MCD = minimal change disease, MN = membranous nephropathy, FSGS = focal segmental glomerulosclerosis, RPGN = rapidly progressive glomerulonephritis).

**Figure 9 biomedicines-11-00365-f009:**
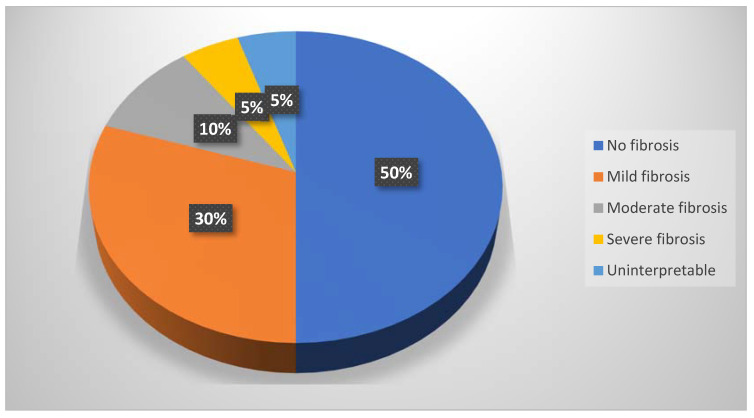
Pie chart of the fibrosis percentages.

**Figure 10 biomedicines-11-00365-f010:**
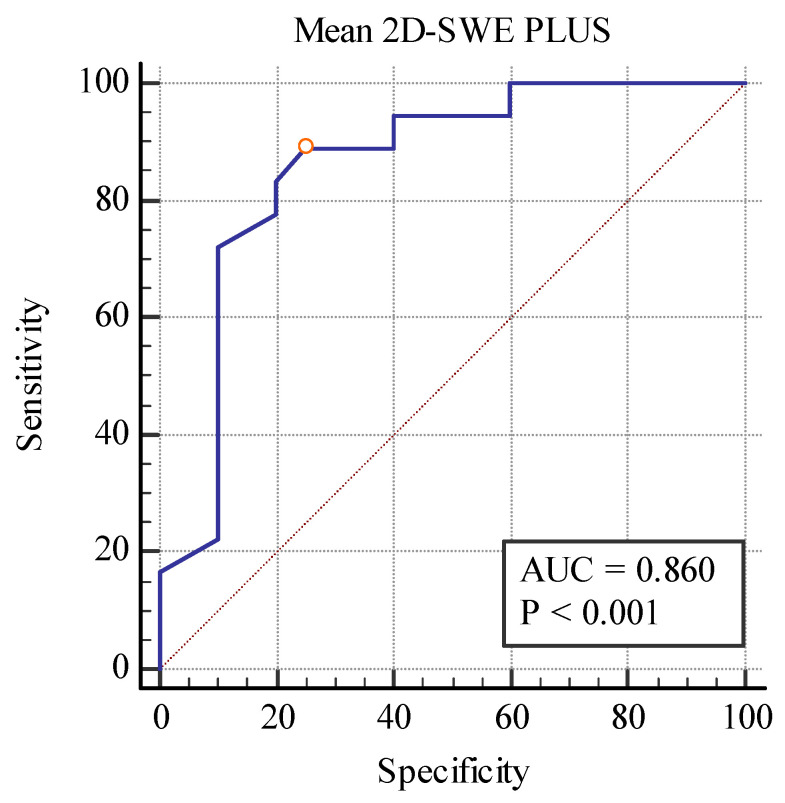
Performance of 2D-SWE PLUS for predicting over 10% fibrosis.

**Figure 11 biomedicines-11-00365-f011:**
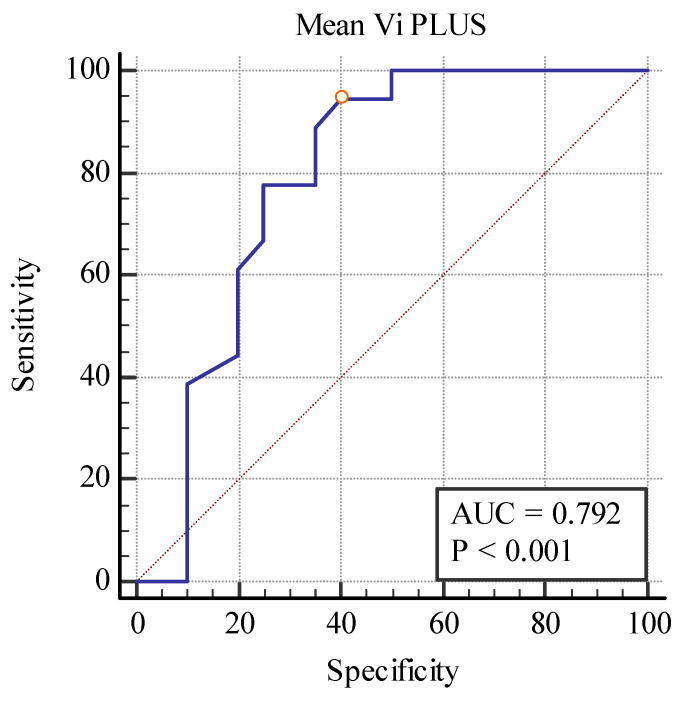
Performance of Vi PLUS for predicting over 10% fibrosis.

**Figure 12 biomedicines-11-00365-f012:**
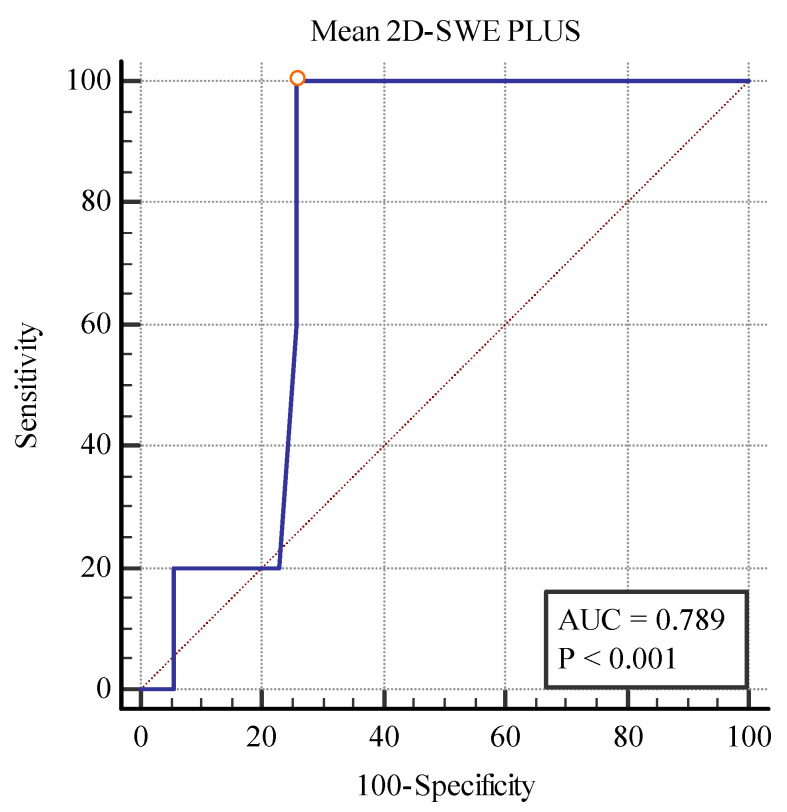
Performance of 2D-SWE PLUS for differentiating between patients with over and under 40% fibrosis.

**Figure 13 biomedicines-11-00365-f013:**
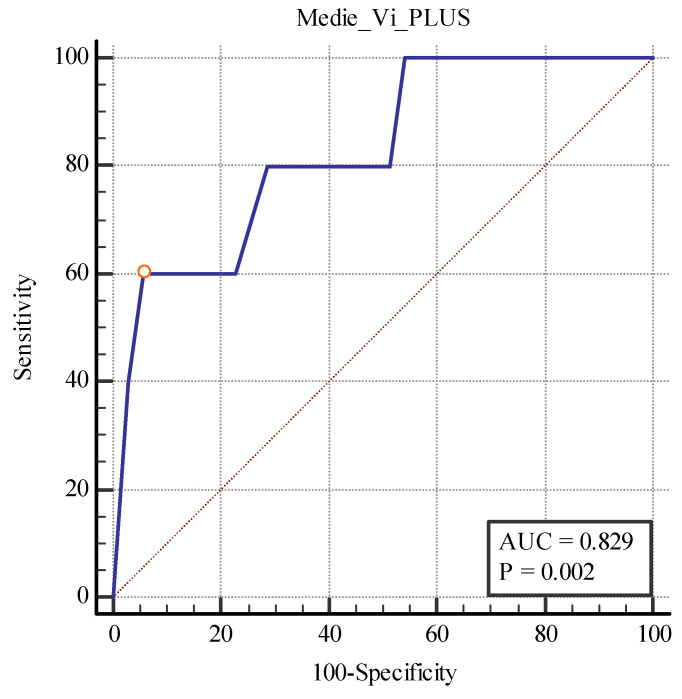
Performance of Vi PLUS for differentiating between patients with over and under 40% fibrosis.

**Table 1 biomedicines-11-00365-t001:** Biological parameters of the patients included in the study.

*N* = 40	Mean Values + SD	Correlation with Kidney Stiffness	Correlation with Kidney Viscosity
Hemoglobin g/dL	12.13 ± 2.7	r = 0.5821, *p* = 0.001	r = 0.1877, *p* = 0.2462
Hematocrit %	36.03 ± 7.8	r = 0.5428, *p* = 0.0003	r = 0.1889, *p* = 0.2429
Serum creatinine mg/dL	2.34 ± 2	r = −0.6569, *p* < 0.0001	r = −0.5282, *p* = 0.0005
Urea mg/dL	76.93 ± 45.94	r = −0.6203, *p* < 0.0001	r = −0.3350, *p* = 0.0346
Proteinuria g/24 h	6.39 ± 7.42	r = −0.1212, *p* = 0.4562	r = −0.2652, *p* = 0.0981
Uric acid mg/dL	6.28 ± 1.38	r = −0.2482, *p* = 0.1225	r = −0.2413, *p* = 0.1336
Total cholesterol mg/dL	189.15 ± 65.66	r = 0.4106, *p* = 0.0085	r = 0.4750, *p* = 0.0020
Triglycerides mg/dL	189.72 ± 107.23	r = 0.3023, *p* = 0.0579	r = 0.4750, *p* = 0.0020
ALAT U/L	24.37 ± 8.03	r = 0.3386, *p* = 0.0326	r = 0.1159, *p* = 0.4762
ASAT U/L	22.92 ± 7.03	r = 0.2015, *p* = 0.2126	r = −0.0896, *p* = 0.5824
Total bilirubin mg/dL	0.53 ± 0.28	r = −0.0390, *p* = 0.8111	r = −0.2535, *p* = 0.1145
Sodium mmol/L	135 ± 21.81	r = 0.09844, *p* = 0.5456	r = 0.2600, *p* = 0.1052
Potassium mmol/L	4.49 ± 0.68	r = −0.04570, *p* = 0.7795	r = 0.04954, *p* = 0.7615
ESR	19.66 ± 19.42	r = −0.04704, *p* = 0.7761	r = −0.07137, *p* = 0.6659
CRP	11.21 ± 31.5	r = −0.2431, *p* = 0.1306	r = −0.3695, *p* = 0.0189

*N* = number of participants, SD = standard deviation, *r* = Pearson correlation coefficient, a *p* value under 0.05 was considered statistically significant.

## Data Availability

Not applicable.

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
