# Peer review of "Relationship between Novel Elastography Techniques and Renal Fibrosis—Preliminary Experience in Patients with Chronic Glomerulonephritis"

_biomedicines, 2023, doi:10.3390/biomedicines11020365_

Round 1

Reviewer 1 Report

The Authors reported their experience in the application of the elestography techniques in patients undergoing  renal biopsy. They found that this method can distinguish between patients with and without renal fibrosis. The use of elastography is already investigate in different settings such as liver and kidney transplant with diverse conclusions.

The population studied in this paper is heterogeneous and most have mild or no fibrosis. I would ask the Authors to compare their results with activity and chronicity index in order to improve the quality of the paper. Indeed, the Authors describe interesting results between inflammation and elastography in different settings and the population studied in this paper (MPGN, LN, etc..) could add further information.

Author Response

We would like to thank the reviewer for the suggestions, which have undoubtedly helped us considerably enhance our paper. We have made major modifications and hope that the publishing of this article will encourage additional research on this topic.

One of the best-defined scores in histopathology is for lupus nephritis but we cannot calculate this activity/chronicity index score given the fact that we have a heterogeneous group of diagnostic entities. Interstitial fibrosis presents itself as a consensus for all of them. We cannot use an activity chronicity index score for patients with renal amyloidosis, patients with minimal change disease, or patients with diabetic nephropathy (where we classify the lesions based on their type and the percent of affected glomeruli). Perhaps a larger number of cases with lupus nephritis would represent a great addition and provide further information. We added this aspect in the limitations section of the manuscript as well.

Reviewer 2 Report

The subject is interesting since it evaluates a non-invasive assessment of kidney function and histopathology. I have the following major comments.

1.      Although fibrosis is a hallmark of CKD, the degree of fibrosis counts in clinical practice. The elastography failed to differentiate the various degrees of fibrosis, but, for instance, 15% fibrosis is very different from 75% fibrosis. This is a severe limitation of the study since the degree of fibrosis has predictive value. Also, it may lead to clinical decisions such as performing or not a biopsy or applying or not a toxic therapy. This point should be clarified, and the need for further improvement of the related technology should be noted.

2.      A regression analysis with the proper covariates should be performed.

3.      The authors notice and discuss the association between CRP and viscosity. They attributed this relation to kidney inflammation. This may or may not happen. A patient with SLE or diabetic complications or simply CKD may have elevated CRP for other reasons. Also, and especially in the lack of multivariate analysis, the CRP relation with viscosity finding deserves the same attention as that of ALAT relation with kidney stiffens.

Author Response

We would like to thank the reviewer for the suggestions, which have undoubtedly helped us considerably enhance our paper. We have made major modifications and hope that the publishing of this article will encourage additional research on this topic.

  1. We would like to thank the reviewer for this observation, we included 2 new AUROCs for differentiating between patients with over or under 40% fibrosis. We would also like to pinpoint the fact that the majority of the patients performed a kidney biopsy with normal or slightly increased serum creatinine and therefore most of them have mild or no fibrosis and only 2 out of 40 have over 50% kidney fibrosis.
  2. Univariate and multivariate regression analyses were provided in the results section, where we included every biological parameter of the patient included in the study, we also modified the text with the pinpointed aspects.
  3. We would like to thank the reviewer for this important remark as well. A patient with SLE or diabetic complications or simply CKD may have elevated CRP for other reasons. We also included this aspect in the text of our manuscript.

Round 2

Reviewer 1 Report

The quality of the paper has improved after revision.

Reviewer 2 Report

The authors address or discuss the raised issues.